# Functional Assessment and Patient-Related Outcomes after Gluteus Maximus Flap Transfer in Patients with Severe Hip Abductor Deficiency

**DOI:** 10.3390/jcm9061823

**Published:** 2020-06-11

**Authors:** Paul Ruckenstuhl, Georgi I. Wassilew, Michael Müller, Christian Hipfl, Matthias Pumberger, Carsten Perka, Sebastian Hardt

**Affiliations:** 1Center for Musculoskeletal Surgery, Charité University Hospital, 10117 Berlin, Germany; michael.mueller@charite.de (M.M.); christian.hipfl@charite.de (C.H.); matthias.pumberger@charite.de (M.P.); carsten.perka@charite.de (C.P.); sebastian.hardt@charite.de (S.H.); 2Department of Orthopaedics and Trauma, Medical University of Graz, 8036 Graz, Austria; 3Department of Orthopaedics, University Hospital Greifswald, 17487 Greifswald, Germany; georgi.wassilew@med.uni-greifswald.de

**Keywords:** gluteus maximus flap transfer, abductor mechanism deficiency, total hip arthroplasty

## Abstract

(1) Background: Degeneration of the hip abductor mechanism, a well-known cause of functional limitation, is difficult to treat and is associated with a reduced health-related quality of life (HRQOL). The gluteus maximus muscle flap is a treatment option to support a severely degenerative modified gluteus medius muscle. Although several reports exist on the clinical outcome, there remains a gap in the literature regarding HRQOL in conjunction with functional results. (2) Methods: The present study consists of 18 patients with a mean age of 64 (53‒79) years, operatively treated with a gluteus maximus flap due to chronic gluteal deficiency. Fifteen (83%) of these patients presented a history of total hip arthroplasty or revision arthroplasty. Pre and postoperative pain, Trendelenburg sign, internal rotation lag sign, trochanteric pain syndrome, the Harris Hip Score (HHS), and abduction strength after Janda (0‒5) were evaluated. Postoperative patient satisfaction and health-related quality of life, according to the Short Form 36 (SF-36), were used as patient-reported outcome measurements (PROMs). Postoperative MRI scans were performed in 13 cases (72%). (3) Results: Local pain decreased from NRS 6.1 (0–10) to 4.9 (0–8) and 44% presented with a negative Trendelenburg sign postoperatively. The overall HHS results (*p* = 0.42) and muscular abduction strength (*p* = 0.32) increased without significance. The postoperative HRQOL reached 46.8 points (31.3–62.6) for the mental component score and 37.1 points (26.9–54.7) for the physical component score. The physical component results presented a high level of positive correlation with HHS scores postoperatively (R = 0.88, *p* < 0.001). Moreover, 72% reported that they would undergo the operative treatment again. The MRI overall showed no significant further loss of muscle volume and no further degeneration of muscular tissue. (4) Conclusions: Along with fair functional results, the patients treated with a gluteus maximus flap transfer presented satisfying long-term PROMs. Given this condition, the gluteus maximus muscle flap transfer is a viable option for selected patients with chronic gluteal deficiency.

## 1. Introduction

Hip abductor deficiency is difficult to treat and leads to pain, limping, and functional limitations [1,2,3]. The most common cause for symptomatic abductor deficiency is intraoperative trauma to the insertion of the gluteus medius (Gmed) and/or minimus (Gmin) muscle at the greater trochanter or to the superior gluteal nerve [2,4]. The consequences include a fatty degeneration of muscular tissue accompanied by restricted function [1,2,3].

Apart from intraoperative reasons, degenerative processes, such as chronic muscle degeneration and tendinopathy, as well as trauma, can lead to lesions of the gluteal muscles [5,6]. Symptoms often appear as greater trochanteric pain syndrome and are associated with trochanteric bursitis and iliotibial band disorders [7,8,9]. Older, socially deprived patients, revision cases, and women are predisposed to these symptoms [1,4].

In addition to conservative treatment options, such as physiotherapy, anti-inflammatory medication, corticosteroids, and lifestyle modification, several surgical techniques are described in the literature. The preferred surgical approach is associated with the size and nature of the muscular lesion [4]. Surgical options include refixation with bone anchors, vastus lateralis advancement, gluteus maximus (Gmax) transfer, and reconstruction with allografts [10]. Broad evidence is lacking, so the need for surgery has to be addressed individually [10,11,12,13]. Nevertheless, recently promising functional results have been published for the Gmax transfer, as described by Whiteside in the case of chronic and large lesions with fatty degenerations of the Gmed and Gmin muscles [4,10,14]. A preoperative, fully functioning Gmax muscle is a necessary prerequisite to performing the Gmax flap procedure [4,10,15].

Patients suffering from abductor mechanism deficiency often present a complex medical history with multiple previous hip operations, predominantly due to infection or implant loosening [4]. The Gmax flap transfer is frequently performed as a salvage procedure due to the limitation of alternative operative treatment options and conservative treatment failure [3,12]. Based on the individual medical history of patients suffering from abductor deficiency, the published studies present inhomogeneous and small cohorts with limitations regarding the compatibility of patients. It can be estimated that patient-related outcome scores, such as health-related quality of life (HRQOL) measurements, should be considered alongside functional results to evaluate treatment success. Moreover, patient-reported outcome measures (PROMs) are increasingly used to quantify the success of surgical hip procedures [16,17].

The purpose of the present study was to examine the success of the Gmax flap transfer for patients suffering from abductor mechanism deficiency. We focus on functional results in conjunction with subjective measurements, such as patient satisfaction and postoperative health-related quality of life. To our knowledge, this study presents one of the largest cohorts of patients treated with a Gmax flap transfer and is the first to consider patients’ postoperative HRQOL.

## 2. Experimental Section

### 2.1. Patients

This retrospective case series consisted initially of all patients surgically treated with a Gmax flap transfer to address abductor mechanism insufficiency between January 2014 and July 2018 in a tertiary referral center. Of all 24 patients, one died, four were lost to follow-up, and one was excluded due to a traumatic femur fracture on the affected extremity. Patients with a suspicion or a history of infection within the last 12 months before the flap transfer were excluded.

Finally, 18 patients with a Gmax flap transfer due to symptomatic abductor insufficiency were included in the present study. The population consisted of 15 females and 3 males, with a mean age of 64 (53‒79) years at the time of surgery. Fifteen of the included patients showed a history of total hip arthroplasty, whereas three patients presented with a severe degeneration of the abductor mechanism without any history of hip replacement. Two of these three patients presented with a history of a joint-preserving surgery. One patient presented with a degenerative abductor deficiency without previous hip surgeries or associated traumas, as shown in Figure 1. Four patients presented with a history of instability after hip replacement with hip dislocation, as shown in Table 1. Before the operation, three patients presented a history of lumbar spine surgery due to a herniated disc. Neurological failure was not detected before or after the operations. Another six patients reported recurrent lower back pain. Other comorbidities were diagnosed, such as hypertension in 12 cases and diabetes mellitus in 5 patients. The mean ASA score regarding the preoperative physical status was 3 (1‒4).

The indication for surgery included the functional limitation of the abductor mechanism, limping, a positive Trendelenburg sign, a positive rotation lag sign, and trochanteric pain syndrome. All patients had been treated conservatively for at least six months without any success.

The fatty degeneration of the Gmed muscle (≥ 2 Goutallier classification) without signs of loosening of the prosthesis were confirmed in preoperative MRI scans and/or X-rays [18]. After the Gmax flap transfer, all patients were immobilized with a unilateral hip cast for six weeks. Crutches with a partial weight-bearing limit of 15 kg were used for early mobilization. The mean time of hospital stay was 9 (7‒16) days. The cast was removed after six weeks and patients were guided step by step towards full weight bearing. Afterwards, patients underwent a standardized stationary rehabilitation program.

### 2.2. Clinical and Functional Evaluation

The Harris Hip Score (HHS), with categories of pain, function, deformity, and range of motion, was recorded pre and postoperatively [19]. Moreover, the strength for abduction was measured after Janda’s classification (0‒5), with 0 indicating no muscular contraction and 5 indicating normal muscle strength [20]. In addition, tenderness on the palpation of the greater trochanter, the Trendelenburg sign, and the internal rotation lag sign were recorded. The Trendelenburg sign was performed in a standing position on one leg. With a drop of contralateral hemipelvis of more than 2 cm, the sign was defined as positive [21,22]. The internal rotation lag sign is a clinical test to indicate abductor muscle lesions. The test was defined as positive if the internal rotation of the hip was not possible or painful in a lateral position of the patient with a flexed knee (45°) and a passively abducted hip joint [23].

### 2.3. Patient-Reported Outcome Measures (PROMs)

To evaluate postoperative patients’ HRQOL, the Short Form 36 questionnaire (SF-36) was performed with the two main categories of Physical (PCS) and Mental Component Summary (MCS) and the eight subcategories, Physical Functioning, Role Physical, Bodily Pain, General Health, Vitality, Social Functioning, Role Emotional, and Mental Health applied [24]. HRQOL scores describe general physical, mental, and social well-being based on the patients’ own evaluations. Patients’ postoperative satisfaction was evaluated on a 1‒5 scale, with 1 indicating not satisfied and 5 indicating fully satisfied. For further subjective evaluation, patients were asked regarding their level of hip pain after the Numeric Rating Scale (NRS 0‒10) and if they would choose this specific surgical treatment again retrospectively [25].

MARS-MRI (Metal Artifact Reduction Sequences—Magnetic resonance imaging) scans were available for radiological evaluation in 13 patients. A musculoskeletal radiologist (GE) performed measurements for muscle volume and muscular degeneration after the Goutallier classification, as shown in Figure 2. The evaluation of all scores and all clinical examinations were performed preoperatively during the admission and postoperatively with specific appointments at the outpatient department of our orthopedic clinic.

### 2.4. Surgical Technique

The Gmax flap was described by Leo Whiteside in 2012 [12]. The described technique is set up to compensate for degeneratively modified Gmed and Gmin muscles, as shown in Figure 3. A preoperative functioning Gmax muscle is mandatory.

The surgical technique, as described by Whiteside, is performed by using a posterior approach to the hip joint [12]. The fascia lata is incised approximately 1.5 cm dorsal to the tensor fascia lata muscle. The gluteus maximus muscle is split in half in length in the direction of the muscle fibers and the incision continues towards the fascia lata below the greater trochanter. The fascia lata is also split in line with its fibers proximal to the iliac crest. This enables the elevation of a triangular muscle flap. A substantial and stable distal fascial flap is very important for the re-attachment to the bone. Afterwards the flap is split into anterior and posterior portions. The posterior flap is positioned and sutured over the femoral neck towards the anterior capsule and the anterior edge of the greater trochanter. This part of the muscle flap is intended to support the small external rotators and provide additional stability. The anterior flap is placed directly on the femur. After preparing the bone with a high-speed surgical drill (Midas Rex Legend EHS Stylus; Medtronic, Dublin, Ireland), to receive a bleeding slot the flap is positioned from the tip of the greater trochanter towards the attachment of the vastus lateralis muscle. The vastus lateralis muscle has been elevated from the proximal insertion and divided in accordance with its fibers. Afterwards the anterior flap is fixed to the femoral slot with transosseous sutures and under the elevated vastus lateralis in a 15° abduction of the leg. The vastus lateralis is then reattached to the distal fibrous tip of the anterior muscle flap. Finally, the lower part of the gluteus maximus is sutured to the fascia lata to cover the anterior flap, as shown in Figure 3 [12].

In the present case series, the published surgical technique for the Gmax flap transfer was marginally modified. A screw was used to additionally stabilize the sinewy part of the anterior flap distal of the greater trochanter, as shown in Figure 3 and Figure 4.

As a result, the force vector of the hip joint movement is transferred towards a posterior direction compared to the native situation [12]. This modification of force transmission is associated with higher shear force and a restriction in hip flexion [26].

Two senior orthopedic surgeons (CP and GW) with more than 10 years’ experience performed all the operations. Surgeries prior to the muscle flap transfer were performed in different orthopedic departments by various surgeons with different surgical approaches and using different implants.

### 2.5. Statistical Analysis

The statistical analysis was performed with IBM SPSS, Version 25, Armonk, NY, IBM Corp. Standard descriptive statistics were used to illustrate all baseline and follow-up parameters. Normally distributed data were presented with the mean and standard deviation; for nonparametric data, the median was calculated. A nonparametric test was performed for data that were not normally distributed. To evaluate the statistical significance, the Wilcoxon and McNemar tests were applied.

The Pearson’s Correlation Coefficient was employed to investigate correlations between variables arising from postoperative SF-36 and HHS outcomes [27]. The correlation coefficient was defined as very high for R-values higher than 0.90, high for values between 0.70 and 0.89, moderate for values between 0.50 and 0.69, low for values between 0.30 and 0.49, and negligible for R-values lower than 0.30.

The level of significance for all tests was set to *p* < 0.05 and all tests were two-sided. A local ethics committee approved the study (Nr: EA4/180/18).

## 3. Results

All 18 patients were examined postoperatively after a mean follow-up of 33.2 (8‒59) months regarding clinical and patient-related outcome measurements. Thirteen patients received a postoperative MRI. None of the four patients with a history of dislocation suffered further dislocation during the follow-up period. One patient showed a single dislocation in the follow-up period without having a history of dislocation. After closed reduction and immobilization for six weeks, no further signs of instability or dislocation were reported. During the follow-up period, no periprosthetic infections were detected.

### 3.1. Clinical and Functional Results

At the time of the postoperative follow-up examination, eight (44%) patients presented with a negative Trendelenburg sign and a negative internal rotation lag sign, whereas 10 (66%) patients presented an unchanged positive test. Four (22%) patients showed postoperative freedom from tenderness on the palpation of the greater trochanter, compared to 14 (88%) who had persistent local pain.

The overall HHS results showed a postoperative improvement from 47.3 (22–94) to 51.1 (26–100) without statistical significance (*p* = 0.345), as shown in Table 2. Postoperatively, the results were excellent for 2 patients, fair in 1 case, and poor for 15 patients. In nine patients the HHS improved, whereas seven showed poorer results and two patients had the same overall condition. The subcategory of “limp” presented a significant postoperative improvement compared to the preoperative results (*p* = 0.003). A history of a slight, moderate, or severe limp was seen for all patients prior to the Gmax flap transfer. After the treatment, no severe limp was recorded, whereas all patients showed a slight or moderate limp. The subcategory of the support decreased significantly (*p* = 0.018). All other subcategories presented a slight improvement, without statistical significance, as shown in Figure 5.

The level of hip pain according to the NRS (0‒10) decreased from 6.1 (0–10) (preoperative) to 4.9 (0–8) (postoperative), without statistical significance (*p* = 0.25). The mean strength of abduction after Janda’s classification showed a nonsignificant increase in strength from 2.7 (1–5) to 2.8 (2–5) (*p* = 0.32), as shown in Table 2. Postoperatively, four patients reached a strength grade of 4 or 5 after Janda, and clinically presented an ability of abduction against resistance on the affected hip joint. Another five patients were able to abduct against gravity (Grade 3). Nine patients reached a postoperative ability of active abduction in a gravity-eliminated position (Grade 2). A level of Grade 1 or 0 was not recorded.

### 3.2. Patient-Related Results

The postoperative HRQOL reached for the Mental Component (MCS) was 46.8 points (31.3–62.6) and for the Physical Component Summary (PCS) it was 37.1 points (26.9–54.7) of the SF-36.

Furthermore, patients reached 29.2 (0–85) for Physical Functioning, 25 (0–100) for Physical Role Functioning, 40.1 (10–100) for Bodily Pain, 42.5 (25–75) for General Health Perceptions, 36.0 (10–55) for Vitality, 65.3 (13–100) for Social Role Function, 38.9 (0–100) for Emotional Role Function and 56.7 (20–84) for Mental Health in the subcategories of the SF-36.

Postoperative satisfaction on a 1‒5 scale reached 3.1 (1–5). One-third of all patients reported complete or almost complete satisfaction (4 or 5). Eight patients reported average satisfaction (3) and four patients reported poor satisfaction or dissatisfaction (1 or 2). Moreover, 13 patients (72%) reported that they would undergo this specific operative procedure again.

A comparison of postoperative HHS and SF-36 results showed a high correlation between PCS and HHS, indicating that patients with good functional scores also presented better results regarding subjective physical function, with statistical significance (R = 0.88, *p* < 0.001). The Mental Component Summary presented a negligible correlation (R = 0.17) with the postoperative HHS results, as shown in Figure 6.

### 3.3. Radiographic Results

The postoperative MRI scans of the abductor mechanism showed, overall, no significant loss of volume. Moreover, no significant changes were found for the level of muscular degeneration according to the Goutallier classification. An intact flap, without signs of flap necrosis, was observed. In some cases, MRI measurements of the muscles were limited due to interfering artifacts. 

## 4. Discussion

The overview of clinical and patient-reported results of 18 patients treated with a Gmax flap transfer due to an abductor deficiency presented diverse results. All surgeries were performed by a surgeon with more than 10 years of experience and an explicit clinical and scientific focus on hip revision surgery. However, the good to excellent clinical results in the published data could not be confirmed in the present study [12,28,29]. Nevertheless, reflecting on the patients’ HRQOL and subjective patient specific outcome measures, the Gmax flap transfer offers satisfying results for the patients affected. This study is the first to present HRQOL results regarding this operative treatment.

Several advantages of the Gmax transfer are reported compared to other surgical techniques [10,12]. It is an appropriate option even in the case of bone loss, as the flap can support a deficient posterior capsule, and the origin as well as the insertion of the advanced muscle is not affected [3,12]. Moreover, it can be effective in the case of THA instability and is recommended for degenerative Gmed tears with a gap extension of up to 10 cm [3,12]. The disadvantages are reported as follows: an experienced surgeon is mandatory; restoration of active abduction is poor; the force vector for abduction is abnormal after the treatment; the results are related to the interposition effect [3,12]. Recent publications presented optimistic functional results for the Gmax transfer that were originally described by Whiteside in 2012 [3,10].

The present study presents fair functional results after Gmax transfer. Overall, no significant clinical improvements regarding the HHS were found postoperatively. In addition, more than half of all patients presented an unaltered positive Trendelenburg sign at the last follow-up. These findings are in contrast to the published results in the literature. The excellent results published in the literature with very small patient cohorts were not reproducible in the present study [12,28,29]. A study by Whiteside and Roy presents a case series of patients who received a Gmax flap transfer during primary hip arthroplasty due to preoperative limping and a positive Trendelenburg sign. Patients with moderate symptoms presented with a negative Trendelenburg sign as well as no pain or limping two and five years postoperatively. Only the group of patients (*n* = 6) with severe avulsion and a degenerative modified abductor mechanism showed similar results. Chandraekaran et al. published a slightly modified technique with a combined Gmax and TFL flap towards the greater trochanter. This case series of three patients with chronic abductor tears presented normal gait and increased abductor strength two years postoperatively [29]. These differences can be explained by a lower preoperative level of muscular degeneration and damage to the abductor. All patients in the present study showed an advanced level of degeneration of the abductor mechanism and a history of multiple prior hip surgeries. Another systematic review indicated functional improvements due to the surgical treatment, depending strongly on the initial amount of muscular damage [4]. This underlines the relevance of the preoperative amount of muscular degeneration. Therefore, it might be recommended to set realistic expectations preoperative, including postoperative limitations for patients with an advanced level of fatty muscular degeneration (≥ 2 Goutallier classification). Nevertheless, the prevention of the further worsening of symptoms and functions can be considered a success, regarding the operative treatment in the present study. A study by Ricciardi et al. treated an inhomogeneous cohort of seven patients with a history of previous hip surgeries with a Gmax flap transfer. Their functional results are similar to the findings of the present study [3]. Along with the fair functional results, the soft tissue coverage was pointed out as a useful protection against pain but also to prevent local seroma or hematoma cavities [3], as shown in Table 3.

Two patients with a history of dislocation presented with a stable THA until the last examination. Postoperatively, these two patients presented with increased postoperative functional and HRQOL results. This is in line with the results of Whiteside et al., who reported a lower rate of dislocation after the Gmax flap transfer and supported the theory of a stabilizing effect towards the posterior capsule through the operative procedure [12]. Nevertheless, one dislocation was seen during the follow-up period in a case without a history of prior dislocation.

Abductor mechanism deficiency is difficult to treat, and various treatment strategies have been suggested in the published literature [4,10]. Although positive results have been reported, patients often still suffer from postoperative functional limitations such as persistent pain, limping, and instability [3,10,12,28]. These complaints are associated with a substantially reduced HRQOL [23]. Therefore, we believe that PROMs are important to quantify the success of the Gmax flap transfer. The Pearson’s correlation coefficient between subjective SF-36 results and objective functional HHS results was high between objective functional results and the subjective Physical Component outcomes. A weak correlation was seen between the mental component and the HHS.

Mental results are influenced due to multiple factors in both directions and therefore may be difficult to connect with the functional outcomes of the surgical procedure. The high correlation between PSC and HHS confirms the subjective level of general physical function with the measured objective results of hip function. As the PCS is set up as a score without a focus towards the affected hip, the positive correlation may underline the importance of good hip function with regard to the general physical condition. Patients with a longer follow-up period tend to show better SF-36 results. Patients may tend to profit from the surgical treatment in the long term. The follow-up period may influence postoperative results. Therefore, short-term realistic expectations have to be discussed with patients preoperatively.

In a case series of three patients, Chandrasekaran et al. reported on PROMs after gluteus flap transfer by using the modified HHS and subjective evaluation regarding patients’ satisfaction on a 0‒10 scale [29]. Two of the three patients reported full satisfaction after the treatment. The present study showed comparable results, whereas only one-third reached full or almost full satisfaction postoperatively. The subjective postoperative satisfaction of the present study is in line with various sets of previous functional and HRQOL results.

The present study included, for the first time, postoperative health-related quality of life results for patients treated with a Gmax transfer flap due to abductor deficiency. Therefore, a comparison with the existing data is limited. Nevertheless, the patient cohort of the present study is facing a similar history of prior surgeries compared to patient cohorts that are suffering from chronic periprosthetic hip infection. The SF-36 results are similar to the results published by Boettner et al. after a two-stage revision for infected total hip arthroplasty [30]. In this study, with a mean follow-up of 61 months, patients reached a physical component score of 36.0 and a mental component score of 45.0 compared to PCS 37.1 and MCS 46.8 in the present study [30]. A literature review of Rietbergen et al. even presents better results (PCS 40.4 and MCS 51.6) after staged hip revision, with 159 patients included and a mean follow-up of 40.7 months [31]. Parvizi et al., in a retrospective study of patients after two-stage revision, reported a mean PCS score of 48.0 and a mean MCS score of 60.0 [32]. These results underline the postoperative limitations and the severity of the abductor mechanism deficiency with respect to the HRQOL. Nevertheless, considering the complex history prior to the gluteus flap procedure and the advanced level of muscular degeneration, realistic patient expectations need to be established. Therefore, the findings regarding patients’ HRQOL support the relevance of the Gmax flap procedure.

Almost three-quarters (72%) of the patients in this study reported that they would choose this surgical procedure again. In general, patients treat an operation as their last resort, after having exhausted more conservative treatment options. Furthermore, the procedure at least seems to prevent the further progression of symptoms. The fact that most patients were mostly satisfied with the procedure also has to be evaluated in the context of the patients’ expectations. All patients were treated conservatively for at least one year without improvement and the surgery was suggested as a salvage procedure. This might have influenced patients’ expectations as well as the PROMs results.

### Limitations

First, the study is limited due to the inhomogeneous study population, which is a result of different medical histories regarding the affected hip joint. Second, a wide range in terms of patients’ ages and follow-up examinations resulted in reduced comparability. Third, the small number of patients (18) is a limitation of the present study. However, larger cohorts of patients for Gmax flap procedures with PROMs do not exist in the literature. 

## 5. Conclusions

The Gmax flap transfer presents diverse functional and PROMs outcomes for patients with chronic abductor mechanism deficiency and an advanced level of muscular degeneration of the Gmed and Gmin muscles. The convincing results published in the literature were not reproducible here. Along with fair functional results, in general there were satisfying patient-reported results. Therefore, the Gmax flap transfer is a treatment option worth considering for patients with a frustrating history of conservative treatment. Nevertheless, severe abductor mechanism deficiency remains difficult to treat, and realistic patient expectations need to be set.

## Figures and Tables

**Figure 1 jcm-09-01823-f001:**
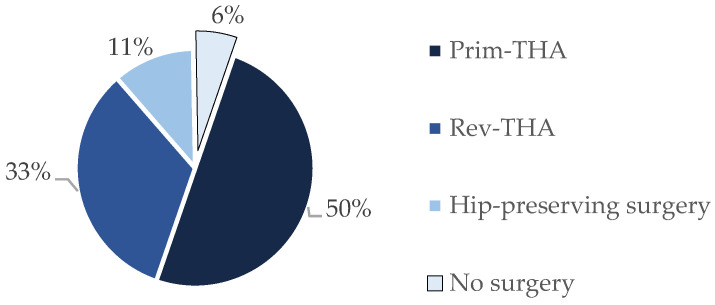
History of surgical treatments prior to the gluteus maximus flap transfer. Prim-THA: Primary Total Hip Arthroplasty, Rev-THA: Revision Total Hip Arthroplasty.

**Figure 2 jcm-09-01823-f002:**
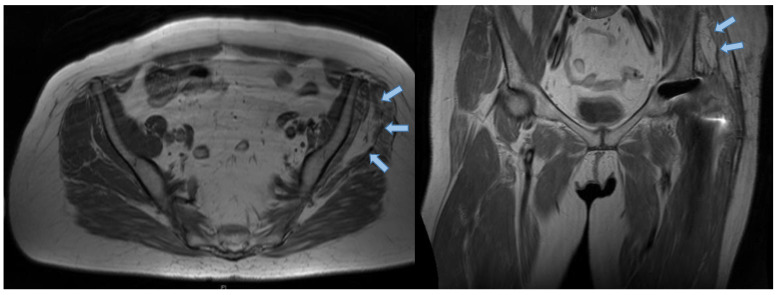
Preoperative MRI scans of a patient suffering from hip abductor insufficiency who presents an advanced level of degeneration of the left gluteus medius (blue arrows) muscle after total hip arthroplasty (Goutallier III). An intact gluteus maximus can be proven. The fatty degeneration of muscular tissue can be seen in T1* weighted MRI scans.

**Figure 3 jcm-09-01823-f003:**
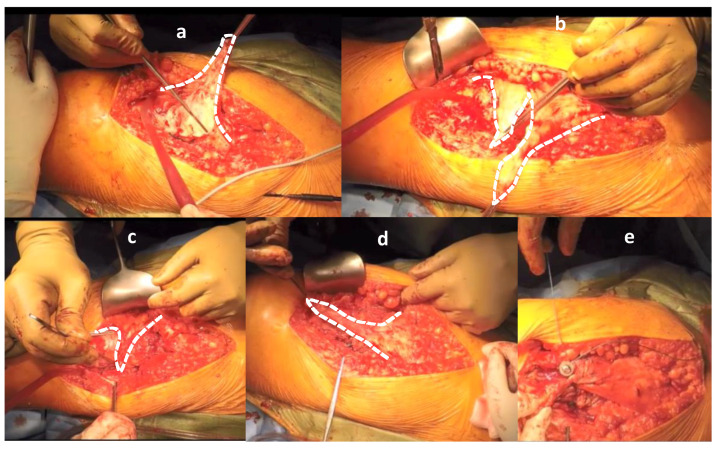
Intraoperative images of the gluteus maximus flap transfer with screw fixation. A part of the gluteus maximus is split, together with a part of the fascia lata in an anterior and posterior portion, and elevated as two flaps (**a**,**b**). The posterior flap is sutured to the anterior part of the capsule to support the gluteus minimus and the capsule (**c**). The anterior part is placed above the posterior flap and sutured to the greater trochanter towards the vastus lateralis (**d**). Additionally, the sinewy part of the anterior flap is fixed with a screw distal to the greater trochanter onto the femur, supported with sutures (**e**). This step, especially, should prevent instability and dislocations [13]. Moreover, the fascia lata is closed over the greater trochanter.

**Figure 4 jcm-09-01823-f004:**
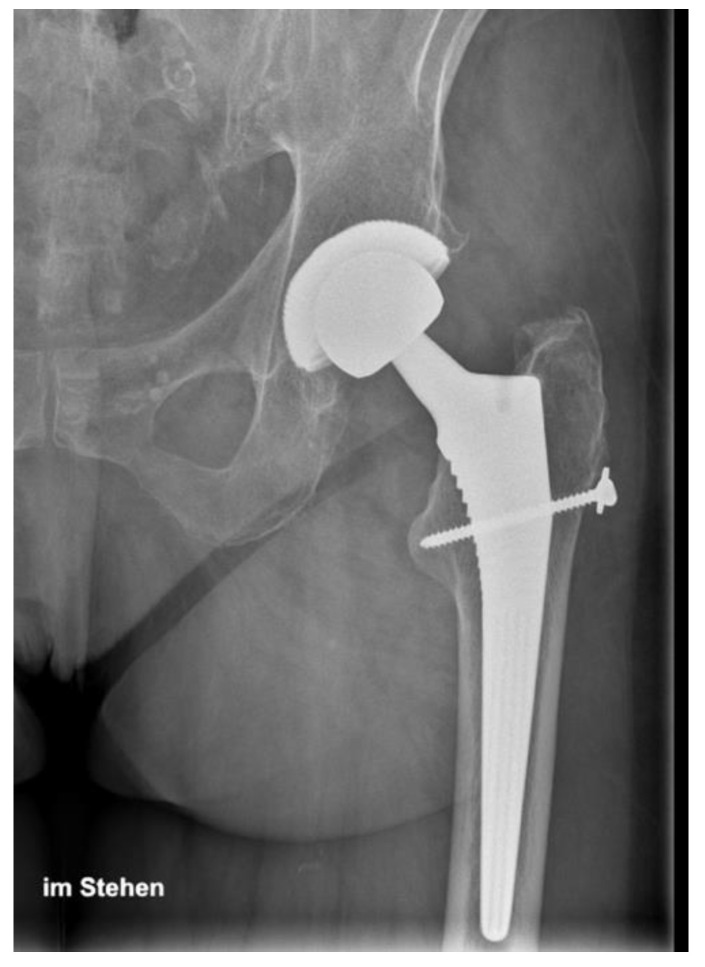
Postoperative X-ray showing the screw fixation of the anterior flap distal of the greater trochanter in the direction of the trochanter minor.

**Figure 5 jcm-09-01823-f005:**
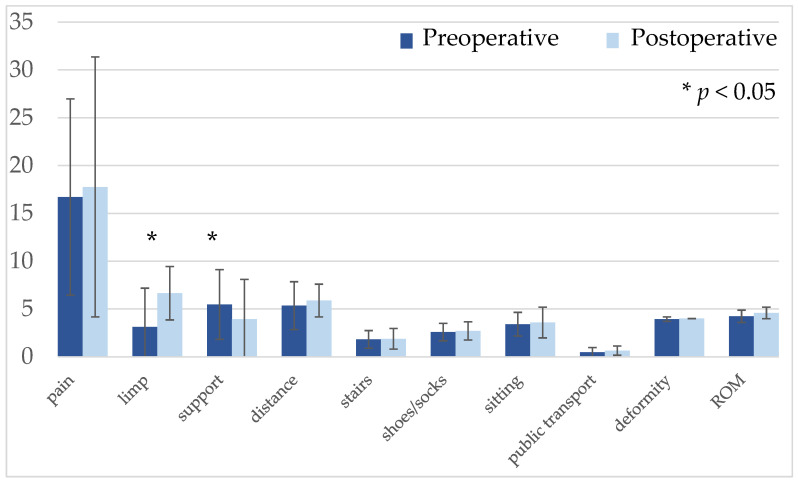
Pre and postoperative HHS results presented significant improvement in the limp category and significantly decreased results for support. ROM: Range of Motion. Statistically significant *p*-value < 0.05.

**Figure 6 jcm-09-01823-f006:**
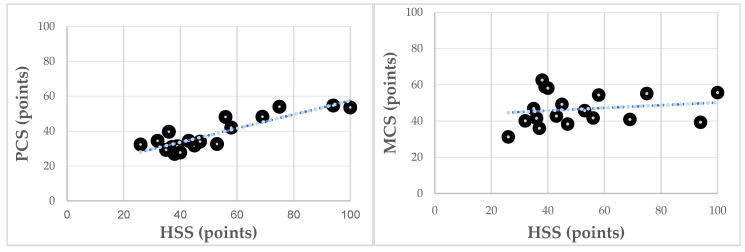
The Pearson’s correlation coefficient presents the correlation of functional results with HRQOL outcomes postoperatively. The main SF-36 categories of Physical Component (PCS) and Mental Component Summary (MCS) were compared with the Harris Hip Score (HHS) results.

**Table 1 jcm-09-01823-t001:** Patient demographics and preoperative baseline characteristics.

Demographics	*n* = 18
Sex M/F, *n* (%)	5 (28%)/13 (72%)
Age (years), mean (range)	64 (53‒79)
BMI (kg/m^2^), mean (range)	26.7 (20–37)
Prior hip surgeries, mean (range)	2.2 (0–5)
Pain (NRS 0‒10), mean (range)	6.1 (0–10)

BMI: Body Mass Index, NRS: Numeric rating scale.

**Table 2 jcm-09-01823-t002:** Functional results, pain, and abduction strength of the affected hip joint pre and postoperative.

	Preoperative	Postoperative	*p*-Value
Harris Hip Score, mean (range)	47.1 (22–94)	51.1 (26–100)	0.42
Pain (NRS), mean (range)	6.1 (0–10)	4.9 (0–8)	0.25
Abduction Strength (Janda), mean (range)	2.7 (1–5)	2.8 (2–5)	0.32

Statistically significant *p*-value < 0.05.

**Table 3 jcm-09-01823-t003:** Overview of studies reporting on the gluteus maximus flap transfer due to chronic abductor mechanism instability.

Authors	Study Design	*n* of Patients	Mean Follow-up (Range)	Clinical Evaluation	PROMs	Radiographic Evaluation (MRI)
Whiteside et al. [13] (2012)	Prospective single center study	11	33 (16‒42)	Trendelenburg Sign, Abduction strength	n.a.	n.a.
Ricciardi et al. [3] (2017)	Retrospective single center study	7	17 (6‒37)	Trendelenburg Sign, Abduction strength	n.a.	n.a.
Chandrasekaran et al. [30] (2017)	Retrospective single center study	3	25 (15‒30)	HOS-SSS, NAHS	mHHS	n.a.

THA: Total Hip Arthroplasty, HOS-SSS: Hip Outcome Score—Sport-Specific Subscale, NAHS: Nonrthritic Hip Score, PROMs: Patient-Reported Outcome Measurements, mHHS: modified Harris Hip Score.

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
