# Peer review of "Functional Assessment and Patient-Related Outcomes after Gluteus Maximus Flap Transfer in Patients with Severe Hip Abductor Deficiency"

_jcm, 2020, doi:10.3390/jcm9061823_

Round 1
Reviewer 1 Report
This review manuscript is interesting, in terms of the restorative technique used which is novel. Perhaps it is a more suitable manuscript for a journal of Orthopedic Surgery because of its specific subject. The first aproximation is the number of the patients. It is apparently low but for this surgical technique it can be considered.
However, there are some observations, that they need to be claryfied.
2.1 Patients
The description of the patients is not clear: In the second paragraph (line 85-89). " 18 patients:15 females and 3 males ". Are there 15 females with a history of total hip arthoplasty and the 3 males who had a history of joint preserving surgery? Is this pathology more common in women? The table 1, need to be revised also.
It is not described the postoperative time to carry out the evaluation of the patients ( Line 99 ) There is diverse interval of follow-up. Was there any specific indication to perform the evaluation? This point is very important to anser HRQOL because the parameters are subjectives and they could change over time.
2.4 Surgical tecnique.
An overly technical description, followed by a serie of images of surgery moments that should be accompanied by some clarifying scheme.
Result and Discussion
3.1 First paragraph They describe the examination of 12 patients results. What happen with the other patients (6) evaluation ?
The results in general do not show great improvements. Although the clinical outcomes are diverse, it seems that the incorporation of HRQOL offers satisfactory results for patients. Perhaps the authors should comment on whether at least this technique can prevent a worsening of the preoperative situation? and not only in their oppinion about "they would choose this surgical procedure again"
Finally, the expertity of the surgeon is very important,In this study two expert surgeons have intervened. Was the variability of the operator taken into account?
References
They need to be review, complete and follow the rules of this journal.
Author Response
Dear Reviewer.
Thank you very much for your careful review and helpful remarks. Please find the point-by-point response attached.
Sincerely,
the Authors

Reviewer 2 Report
This article is interesting and presents an attractive technique.
The authors present an insufficient size in relation to their conclusions. The study is descriptive, and is too preliminary for this journal.
The surgical technique is insufficiently described. I would advise the authors to include more self-explanatory diagrams of their technique.
Statistical methods must be sufficiently described, and the statistical package used must be indicated. Values ​​should be expressed as IQR.
Table 1 of the description of the patients is very cumbersome and not very descriptive. Figure 1 should be better described.
Figure 4 is very basic, it must express the error bars, and the statistical significance is not known what it refers to. Describe acronyms in the figure footer.
The results are promising, but very brief for the reach of this important journal. Other aspects of patients such as comorbidities should be indicated. More emphasis should be placed on gender and age differences. Longer follow-up of patients should be done over time.
The discussion is brief. Authors should review the way of naming the bibliography.
Although the limitations of the study are justified by the authors, I would advise the authors to include a table where the most relevant articles of their technique are collected to give it greater relevance.
Author Response

(The authors gave the same response as above.)

Reviewer 3 Report
I think this is a well written article and there are little problems as investigation. These are very difficult cases and, as a result, were slightly challenging, I suppose. However, proper operations were carried out under the close evaluations in this case series. The number of cases is relatively abundant and it is a considerably a beneficial report for readers about the believability of the result. About the following points, I would like to have you add considerations.
From this result, the group which was not improved is mixed with an improved group in various measurement indexes. About the difference of each group, I would like you to add consideration about their causes.
As this operation performs recovery of the muscular strength of abductors for a unitary purpose particularly, I would like you to add consideration based on a result well about the point. After that, it will be necessary to clarify whether this operation method is useful for what kind of patient.
In addition, please confirm the following points and correct or add comments.
Line 178-18-
As the number of totals does not match, please revise it.
Line 219-221
You should show a little more objective and qualitative results including the method. Did the Goutallier classification change? How much did the area become large or small?
Author Response

(The authors gave the same response as above.)

Round 2
Reviewer 1 Report
The authors have responded satisfactorily to my concerns.
I therefore believe that it can be considered for publication.
Author Response
Reviewers' comments:
Reviewer #1:
The authors have responded satisfactorily to my concerns.
I therefore believe that it can be considered for publication.
- Author response:
Thank you very much for your constructive review and helpful remarks during the review process. Due to minor grammatical concerns of another reviewer, a professional English correction was performed by using the journals language-editing tool.
Sincerely,
The authors
Reviewer 2 Report
The authors have carried out an exhaustive review of the document, they have answered the questions presented. The graphics should be in colors to be more striking and visual for the reader. Numerical results should be expressed as mean [IQR].
Authors should review their grammar and their writing form for final acceptance.
Author Response
Reviewers' comments:
Reviewer #2:
The authors have carried out an exhaustive review of the document, they have answered the questions presented. The graphics should be in colors to be more striking and visual for the reader. Numerical results should be expressed as mean [IQR].
- Author response I:
Thank you very much for your constructive review and helpful remarks during the review process. The graphics were highlighted with colors to be more striking for the reader. Furthermore, numerical results are expressed as mean [range] in the revised version of the manuscript.
Authors should review their grammar and their writing form for final acceptance.
- Author response II:
A professional English correction was performed by using the journals language-editing tool. The certificate is attached.
Sincerely,
The authors

Reviewer 3 Report
Almost that I pointed out are improved particularly, the authors need not revise it further.
Author Response
Reviewers' comments:
Reviewer #3:
Almost that I pointed out are improved particularly, the authors need not revise it further.
- Author response:
Thank you very much for your constructive review and helpful remarks during the review process. Due to minor grammatical concerns of another reviewer, a professional English correction was performed by using the journals language-editing tool.
Sincerely,
The authors